# Adherence to Mediterranean Diet in Individuals on Renal Replacement Therapy

**DOI:** 10.3390/ijerph20054040

**Published:** 2023-02-24

**Authors:** Elisabetta Falbo, Gabriele Porchetti, Caterina Conte, Maria Grazia Tarsitano

**Affiliations:** 1Azienda Sanitaria Provinciale di Catanzaro, Centro di Medicina del Viaggiatore e delle Migrazioni, 88100 Catanzaro, Italy; 2Department of Human Sciences and Promotion of the Quality of Life, San Raffaele Roma Open University, Via di Val Cannuta 247, 00166 Rome, Italy; 3Department of Endocrinology, Nutrition and Metabolic Diseases, IRCCS MultiMedica, Via Milanese 300, Sesto San Giovanni, 20900 Milan, Italy; 4Department of Medical and Surgical Science Nutrition Unit, University Magna Grecia, 88100 Catanzaro, Italy

**Keywords:** dialysis, kidney transplant, Mediterranean diet

## Abstract

Patients on renal replacement therapy are typically subject to several dietary restrictions; however, this approach has been questioned in recent years, with some suggesting that the Mediterranean diet might be beneficial. Data on the adherence to this diet and factors that influence it are scarce. We conducted a web survey among individuals on renal replacement therapy (dialysis or kidney transplant, KT) using the MEDI-LITE questionnaire to assess adherence to the Mediterranean diet and dietary habits in this population. Adherence to the Mediterranean diet was generally low, and significantly lower among participants on dialysis versus KT recipients (19.4% vs. 44.7%, *p* < 0.001). Being on dialysis, adopting fluid restrictions, and having a basic level of education were predictors of low adherence to the Mediterranean diet. Consumption of foods typically included in the Mediterranean diet, including fruit, legumes, fish, and vegetables, was generally low, particularly among those on dialysis. There is a need for strategies to improve both the adherence to and the quality of the diet among individuals on renal replacement therapy. This should be a shared responsibility between registered dietitians, physicians, and the patient.

## 1. Introduction

Patients on renal replacement therapy, particularly those on dialysis, are typically subject to several dietary restrictions [1]. Currently available guidelines provide general recommendations on energy intake (e.g., 30–35 kcal/kg body weight), protein (1–1.2 g/kg/day), potassium (1500–2000 mg/day if elevated), phosphorus (800–1000 mg/day if elevated), and sodium intake (<100 mmol/day or <2.3 g/day) and suggest encouraging diets meeting the recommended dietary allowance for adequate intake of all vitamins and minerals [1]; however, they do not recommend specific dietary patterns due to lack of randomized clinical trials in this setting. Use of the Mediterranean diet may be feared by physicians because of its high content in fruits, vegetables, and legumes, which may contain relevant amounts of potassium and phosphate. Dietary restrictions may sometimes be overzealous, leading to deficiencies in important nutrients and poor nutritional status [2]. However, in recent years, this approach has been questioned, both due to the relative lack of evidence supporting such restrictions and to the potential nutritional deficiencies and deprivation of beneficial dietary elements associated with strict dietary regimens [2,3]. The Mediterranean diet is a common eating pattern found in Italy, Greece, Spain, and other countries in the Mediterranean basin. This pattern encompasses nutritious eating habits characterized by a high intake of vegetables, fresh fruits, legumes, and cereals, which serve as major sources of fiber and antioxidants, as well as a moderate intake of alcohol. Fish, nuts, and olive oil also ensure a high intake of mono-unsaturated fatty acids [4]. Adherence to the Mediterranean diet clusters with other healthy behaviors, including physical exercise and abstention from smoking, and is related to the level of education [5] and age [6]. In Italy, the highest adherence has been reported in the central regions, whereas people living in the north seem to have the lowest adherence [7]. The Mediterranean diet was shown to reduce cardiovascular risk [8] and to help prevent [9] or delay the progression [10] of chronic kidney disease (CKD). At present, the National Kidney Foundation’s Kidney Disease Outcomes Quality Initiative (KDOQI) Clinical Practice Guideline for Nutrition in CKD suggests the Mediterranean diet for adults with CKD stage 1–5 not on dialysis or post-transplantation to improve lipid profile [1]. Data on patients on dialysis or kidney transplant (KT) recipients are scarce. The available evidence indicates that in patients on maintenance dialysis, greater adherence to the Mediterranean diet is associated with better cardiac geometry [11] and reduced all-cause mortality [12], although other studies suggest that there is no effect on cardiovascular or total mortality in patients on hemodialysis [13]. In KT recipients, greater adherence to the Mediterranean diet has been associated with lower risk of metabolic syndrome [14], post-transplant diabetes [15], graft failure [16], and mortality [17]. Assessing adherence to the Mediterranean diet, identifying factors that affect it, and detecting potential unhealthy food consumption patterns is therefore important in improving the nutritional care of patients on renal replacement therapy. We sought to assess adherence to the Mediterranean diet among individuals on dialysis or who received a KT and to investigate factors that influence it.

## 2. Materials and Methods

### 2.1. Study Design

This was a secondary analysis of a cross-sectional, observational study aimed at providing information on the burden of obesity and the lifestyle habits of patients on renal replacement therapy in a real-life setting [18]. Data were obtained from an anonymous, open online survey among adult (age ≥ 18 years) individuals on renal replacement therapy (KT or dialysis). The study design has been previously described [18]. Briefly, the survey was published on a dedicated web page on Google Forms between 1 January 2022 and 31 March 2022 and was advertised on social media platforms by the main national associations of patients with polycystic kidney disease or those on renal replacement therapy. All participants who accessed the informed consent page and consented to participate were included in this study. We collected, among other variables, self-reported height and weight to compute body mass index (BMI), age, sex, smoking status, weight changes after KT, and dietary habits (as detailed in the following paragraph). The total number of questionnaire items was 45 on a single web page. Only questionnaires with complete answers were analyzed.

### 2.2. Adherence to the Mediterranean Diet

Adherence to the Mediterranean diet and dietary habits were assessed using the MEDI-LITE questionnaire, a validated questionnaire that assesses the consumption of nine food categories (fruit, vegetables, legumes, cereals, fish, meat and meat products, dairy products, alcohol, and olive oil) [19,20]. The questionnaire is validated for use in the Italian population and yields a score ranging from 0 (minimum) to 18 (maximum). For foods typically part of a Mediterranean diet (fruit, vegetables, cereals, legumes, and fish), 2 points are assigned to the highest (optimal) category of consumption, 1 to the intermediate category, and 0 to the lowest category. For foods not typically part of a MD (meat and meat products, dairy products), 2 points are assigned to the lowest (optimal) category, 1 to the intermediate category, and 0 to the highest consumption category. For alcohol, 2 points are assigned to the middle category (optimal, 1–2 alcohol units/day), 1 to the lowest category (1 alcohol unit/day), and 0 to the highest category (>2 alcohol units/day) of consumption [19]. For olive oil, 2 points are assigned for regular (optimal) use, 1 for frequent use, and 0 for occasional use. Adherence to the Mediterranean diet was considered adequate if the score was >9 [21,22].

### 2.3. Physical Activity

The level of physical activity was assessed using the International Physical Activity Questionnaire (IPAQ) short form, which estimates the level of physical activity (low, moderate, high) based on the subject-reported activities (vigorous/moderate physical activity and walking) relative to the 7 days prior to completion of the questionnaire and is validated for use in the Italian population [23].

### 2.4. Ethical Approval

The study was carried out in accordance with the Declaration of Helsinki and approved by the Ethics Committee of IRCCS San Raffaele Roma (ODIRT protocol, nr. 21/29). The voluntary questionnaire could be completed only by participants who provided their informed consent to be enrolled in the study. The questionnaire was anonymous, no information that could render the data subject identifiable was collected. The results are reported according to the Checklist for Reporting Results of Internet E-Surveys (CHERRIES) [24]

### 2.5. Statistical Analysis

Descriptive statistics were obtained for all study variables. Normality was assessed with the Shapiro–Wilk test. Continuous variables were expressed as mean ± standard deviation or median (25th–75th percentile), depending on the data distribution. Categorical variables were summarized as counts and percentages. The Mann–Whitney U test or the Kruskal–Wallis test were used for between-group comparisons for continuous variables. The Fisher exact test or the χ^2^ test was used to assess the association between categorical variables. Univariable and multivariable logistic regression (including all significant variables at univariable regression) models were used to identify variables associated with inadequate adherence to the Mediterranean diet, adjusting for age and sex. All variables were screened for violations of the assumptions relevant to each of the statistical analyses performed. Multicollinearity was assessed by calculating the variance inflation factors (VIFs, with a threshold of 5). Statistical significance was set at *p* < 0.05. Statistical analysis was conducted using IBM SPSS Statistics (IBM SPSS Statistics for Windows, Version 28.0., IBM Corp.: Armonk, NY, USA).

## 3. Results

A total of 333 potential participants accessed the informed consent page, of whom 322 consented to participate in the study and were included in the present analysis. Participant characteristics are presented in Table 1.

The majority of participants were male, median age was 56.0 (48.0; 62.0) vs. 54.0 (45.0; 62.0) years in KT recipients and participants on dialysis, respectively; *p* = 0.33. The median BMI was higher in KT recipients than in participants on dialysis (23.9 (21.6; 26.5) vs. 23.6 (20.7; 27.1) kg/m^2^, respectively; *p* < 0.001), with numerically higher proportions of participants with underweight or obesity in the latter group (Table 1). Overall, 44.1% of participants reported having gained weight since the start of dialysis or since KT, the proportion being significantly higher among KT recipients (Figure 1).

In both groups, the majority of participants lived in Northern Italy, had an intermediate level of education (upper secondary or post-secondary non-tertiary education [25]), and were workers. The proportion of participants with comorbidities was similar between the groups, with numerically higher figures for hypertension and dyslipidemia in the KT group. Median time since KT was 6.0 (3.0; 13.0) years. Approximately half (52%) of KT recipients were on steroid therapy.

### 3.1. Adherence to the Mediterranean Diet

The median MEDI-LITE score was 9.0 (8.0; 10.0) in KT recipients and 8.0 (7.0; 9.0) in participants on dialysis (*p* = 0.001). The proportion of participants with adequate adherence to the Mediterranean diet (MEDI-LITE score > 9) was significantly higher among KT recipients as compared with participants on dialysis (44.7% vs. 19.4%, *p* < 0.001). There were no differences in the proportion of participants with comorbidities (diabetes, hypertension, dyslipidemia, vascular disease, obesity) between those with inadequate and adequate adherence to the Mediterranean diet. Overall, 17.4%, 29.2%, 56.2%, and 19.6% of participants reported having to restrict the intake of fluids, potassium, salt, or phosphate, respectively. These restrictions were significantly more common among participants on dialysis (Figure 2).

In participants on dialysis, the MEDI-LITE score did not differ among individuals of different sex (*p* = 0.590), age (*p* = 0.715), or BMI (*p* = 0.901) categories, from northern, central, or southern Italy (*p* = 0.408), by the level of education (*p* = 0.103), working status (*p* = 0.296), or level of physical activity (*p* = 0.266) (Figure 3).

In KT recipients, the MEDI-LITE score was significantly lower in participants living in southern Italy as compared with those living in central Italy (*p* = 0.031) (Figure 4). Participants with a basic level of education had the lowest adherence to the Mediterranean diet, with significant differences as compared both to those with an intermediate and advanced level of education (*p* = 0.045, Figure 4).

There was no difference between sexes (*p* = 0.640), among age (*p* = 0.715) or BMI (*p* = 0.514) categories, by working status (*p* = 0.633), or the level of physical activity (*p* = 0.382).

At univariable logistic regression in the whole population, a basic level of education was associated with a nearly 3-fold increase in the odds of inadequate adherence to the Mediterranean diet (OR 2.72, 95% CI (1.27; 5.86); *p* = 0.01). Dietary limitations such as restricting the intake of fluids (OR 4.14, 95% CI (1.20; 8.84); *p* < 0.001), potassium (OR 2.23, 95% CI (1.32; 3.78; *p* = 0.003)), or phosphate (OR 3.39, 95% CI (1.73; 6.67; *p* < 0.001)) were also predictors of inadequate adherence to the Mediterranean diet. No other variables (sex, age, BMI, geographical area, working status, physical activity level, salt restriction) were associated with inadequate adherence to the Mediterranean diet. Assessment of collinearity revealed a VIF = 5.73 for being on dialysis. Therefore, two separate multivariable regression analyses were conducted to assess the role of being on dialysis and to investigate the effect of specific dietary restrictions. At multivariable logistic regression, basic level of education, being on dialysis (Table 2), and fluid restriction retained significance (Table 3).

### 3.2. Consumption of Specific Food Categories

To provide further insight into the dietary habits of individuals on renal replacement therapy, we investigated the proportion of participants scoring 2 (optimal), 1 (intermediate), or 0 (inadequate) on the consumption of the food categories included in the MEDI-LITE questionnaire (Figure 5).

#### 3.2.1. Fruit

Fruit consumption patterns were similar between KT recipients and participants on dialysis, with nearly half of them scoring 0 (less than one serving per day) and no significant differences between groups. There were no differences in the proportion of participants with comorbidities (diabetes, hypertension, dyslipidemia, vascular disease, obesity) between those with inadequate fruit consumption and other participants.

#### 3.2.2. Vegetables 

A significantly greater proportion of participants on dialysis reported consuming low amounts of vegetables (less than one serving per day) as compared with KT recipients, most of whom reported an intermediate consumption of vegetables. There were no differences in the proportion of participants with comorbidities between those with inadequate vegetable consumption and other participants.

#### 3.2.3. Legumes

Significant differences were identified between groups with regard to legume consumption. Participants on dialysis reported consuming significantly lower amounts of legumes than KT recipients. However, even in the latter group, more than half of the participants reported a low consumption of legumes (less than one serving per week). There were no differences in the proportion of participants with comorbidities between those with inadequate legume consumption and other participants.

#### 3.2.4. Cereals

There were no significant differences in the consumption of cereals between groups. In both groups, approximately 40% of the participants reported consuming less than one serving of cereals per day (0 points). There were no differences in the proportion of participants with comorbidities between those with inadequate cereal consumption and other participants.

#### 3.2.5. Fish

The proportion of participants consuming an optimal amount of fish (more than 2.5 servings per week) was similarly low in the two groups. Approximately half of the participants in both groups reported eating less than one serving per week. There were no differences in the proportion of participants with comorbidities between those with inadequate fish consumption and other participants. 

#### 3.2.6. Meat and Meat Products

The proportion of participants with optimal consumption of meat and meat products, including cured meats (less than one per day), was numerically higher in the KT group. Approximately 15% of the participants in each group reported eating more than 1.5 servings of meat per day. The proportion of participants with obesity was significantly greater among those with excess meat consumption than in other participants (18.8% vs. 9.1%, *p* = 0.045).

#### 3.2.7. Milk and Dairy Products

A high proportion of participants reported consuming less than one serving of milk or dairy products per day. This figure was significantly higher in participants on dialysis, whereas the proportion of participants who reported having more than 1.5 servings per day was significantly greater among KT recipients. The proportion of participants with diabetes was significantly greater among those with excess consumption of dairy products than in other participants (33.3% vs. 17.0%, *p* = 0.014).

#### 3.2.8. Alcohol

Most participants reported drinking less than one alcohol unit per day, this proportion being significantly higher in the dialysis group. No participants in the dialysis group reported inadequate alcohol consumption (more than 2 alcohol units per day), as opposed to 2% in the KT group. The proportion of participants who consumed 1–2 alcohol units per day was numerically higher in the KT group. The proportion of participants with diabetes was significantly greater among those with excess alcohol consumption than in other participants (60% vs. 18.3%, *p* = 0.018).

#### 3.2.9. Olive Oil

The proportion of participants who reported regular consumption of olive oil was similarly high in the two groups. Only 7.5% and 12.9% of participants in the dialysis and KT group, respectively, reported using olive oil only occasionally. The proportion of participants with diabetes (31.6% vs. 17.3%, *p* = 0.034) or obesity (21.1% vs. 9.2%, *p* = 0.043) was significantly higher among those with inadequate olive oil consumption than among other participants.

## 4. Discussion

In the present study, we assessed the adherence of individuals on renal replacement therapy to the Mediterranean diet, and investigated what factors influence it. Adherence to the Mediterranean diet was generally lower (median Medi-Lite score 9.0 in KT recipients and 8.0 in participants on dialysis) than in the general Italian population, where the reported median Medi-Lite score is 12 [19]. To the best of our knowledge, no studies used the Medi-Lite questionnaire to assess adherence to the Mediterranean diet in CKD patients 1–5 not on renal replacement therapy. Using the Mediterranean diet serving score, Bučan Nenadić and colleagues found that as low as 9.1% of participants with CKD were adherent to the Mediterranean diet, which is lower than the 44.7% and 19.4% we found in KT recipients and patients on dialysis. The finding that the adherence was lowest in participants on dialysis was, at least in part, due to dietary restrictions, which were significantly more common among participants on dialysis. In fact, being on dialysis increased the likelihood of poor adherence to the Mediterranean diet, possibly due to the need of restricting fluid intake. Basic education (primary or lower secondary education) was also a factor significantly associated with the increased likelihood of inadequate adherence to the Mediterranean diet. The latter association has also been reported in the general Italian population [19]. The association between a lower level of education and unhealthy dietary patterns has been reported in several large studies conducted in different countries [26,27,28] and highlights the need for large-scale educational programs aimed at increasing nutrition literacy, especially among individuals with a lower level of education. Greater nutrition literacy is in fact associated with greater adherence to healthy dietary patterns [29], even in kidney transplant recipients [30], and specifically designed educational programs have been shown to improve adherence to the Mediterranean diet in vulnerable individuals living in extreme poverty [31].

The lack of association with a Geographical area is in contrast with previous studies in the general population showing the lowest adherence and consumption of fruit and vegetables in the Northern regions of Italy [7,32]. This discrepancy might be due to a relatively greater proportion of participants from Northern Italy and the generally low adherence to the Mediterranean diet in our study.

Our finding of low adherence to the Mediterranean diet is consistent with the few previous studies conducted in patients on hemodialysis [13] or KT [16], indicating that only a small proportion of patients on renal replacement therapy follow a Mediterranean diet pattern. According to current recommendations, individuals with CKD including those post-transplantation should modulate their dietary intake with the aim of maintaining levels of potassium and phosphate in the normal range [1]. However, very strict dietary limitations are often imposed on these patients, whose appropriateness has been sometimes called into question [2]. It is not surprising that participants on dialysis had the lowest adherence to the Mediterranean diet, which is rich in fruits, vegetables, and legumes that may be relevant sources of potassium and phosphate. Guidelines state that it is reasonable to adjust the dietary intake of these minerals to maintain serum levels within the normal range [1]. This possibly explains why, in our study, adherence to the Mediterranean diet was influenced by dietary restrictions typically adopted in this population.

The benefits and potential risks of adopting the Mediterranean diet in the CKD population have recently been reviewed by the European Renal Nutrition (ERN) Working Group of the European Renal Association–European Dialysis Transplant Association (ERA-EDTA) [33]. The Mediterranean diet is characterized by high consumption of fruits and vegetables, which are high in potassium, and concerns exist that it could predispose CKD patients to hyperkalemia and acid/electrolyte disbalance. Our findings indicate that dietary restrictions such as limiting potassium or phosphate intake and, above all, limiting fluid intake were associated with reduced adherence to the Mediterranean diet. However, the predominance of plant and fish protein versus meat protein might decrease phosphate bioavailability, thus reducing the phosphate load [33]. Regarding dietary potassium, it is not yet clear whether it increases the risk of hyperkalemia, as intracellular/extracellular potassium shifts may be influenced by several factors, including acid–base balance and medications [34]. Furthermore, in patients with end-stage renal disease whose kidneys cannot handle a potassium load, the gut plays an important role [35,36], and constipation could favor hyperkalemia. In this light, providing an adequate amount of fiber as recommended in the Mediterranean diet might help ensure proper bowel movements to counterbalance a relative increase in dietary potassium. However, until more evidence is available in support of or against increasing the consumption of fruit and vegetables in patients at risk of hyperkalemia, choosing fruits and vegetables with low potassium content, implementing strategies to reduce the potassium content in these foods such as leaching vegetables, and careful monitoring of potassium serum levels might be more prudent.

Fluid retention is associated with worse health outcomes in dialysis patients [37,38]; therefore, limiting fluid intake is recommended. However, it has been suggested that the benefit of fluid restriction is achieved only if nutritional status and food intake are adequate [2,37]. In our analysis, fluid restriction, which was adopted almost exclusively by participants on dialysis, was an independent predictor of inadequate adherence to the Mediterranean diet, indicating low consumption of foods characterizing this dietary pattern (mainly fruit, vegetables, legumes, and fish). As discussed further in the following paragraphs, this might lead to a reduced intake of beneficial nutrients.

The analysis of specific food categories revealed that consumption of fruit, vegetables, and legumes was generally inadequate with respect to the Mediterranean diet model, whereas most participants reported optimal consumption of dairy products and olive oil. The use of olive oil, which is a mainstay of the Mediterranean diet, should be encouraged, as it may help counteract constipation in patients on hemodialysis [39], and it exerts anti-inflammatory and anti-atherogenic effects that may be particularly beneficial in patients at high cardiovascular risk such as patients on renal replacement therapy [40].

Dairy products are a source of potassium and phosphorus [1]. However, they also provide calcium, α-linolenic acid, and other heart-healthy nutrients [1,41]. Very little/no information is available on the effects of dairy consumption on health outcomes in patients on renal replacement therapy. A study in CKD patients not on dialysis nor transplanted found that dairy consumption was associated with lower renal function [42]. In the general population, the consumption of dairy is associated with better renal health [43], reduced risk of overweight or obesity (especially milk and yogurt), hypertension (low-fat dairy and milk), and type 2 diabetes (yogurt) [44]. We found that excess consumption of dairy products was significantly associated with having diabetes. It is possible that higher rates of diabetes were related to increased intake of high-fat dairy. Unfortunately, we were not able to discriminate between high-fat and low-fat dairy products.

A high proportion of participants consumed an inadequate amount of fish. Fish is a natural source of long-chain omega-3 polyunsaturated fatty acids, which are known for their cardioprotective properties [45]. Current guidelines on renal nutrition suggest prescribing omega-3 polyunsaturated fatty acids to reduce triglycerides and LDL cholesterol and raise HDL cholesterol levels but do not recommend routine use of these compounds to reduce mortality or cardiovascular risk [1]. To the best of our knowledge, the relationship between fish consumption and health outcomes has not been explored in patients on renal replacement therapy, but it is likely that eating fish carries the same benefits as taking fish oil supplements. Wild (as opposed to farmed) sardine, mackerel, salmon, and other high-content marine-based foods should be chosen to increase blood/tissue levels of eicosapentaenoic and docosahexaenoic acid, although achieving high daily intake may be challenging [1]. Lastly, as already mentioned, fish protein might be more advantageous with regard to the phosphate load as compared to meat protein [33].

When we compared participants on dialysis with KT participants, we found that the consumption of vegetables, legumes, and dairy products was lower in those on dialysis. In a study assessing the association between Mediterranean diet and cardiac geometry in patients with CKD, the consumption of specific food categories, such as vegetables and legumes, was associated with a more favorable cardiac remodeling pattern [11]. According to recent recommendations, decisions about phosphate restriction should consider the bioavailability of phosphorus sources. The phosphate content of vegetables and legumes has low biological availability [46]. Therefore, moderate consumption of these should be considered even for patients on dialysis. Furthermore, vegetable intake has been associated with a reduced risk of post-transplantation diabetes [47]. In a large prospective cohort study of cardiovascular and all-cause mortality, the effect of the Mediterranean diet or dietary approaches to stop hypertension (DASH) on the risk of cardiovascular or all-cause mortality was neutral [13], suggesting that dietary patterns that are beneficial in the general population do not have the same effect in patients on hemodialysis. It is also possible that only some food categories have beneficial effects. In the same cohort, increasing the consumption of fruits and vegetables to approximately 2 to 3 servings per day was associated with a 20% lower risk of all-cause mortality and death from non-cardiovascular causes [48]. These findings indicate that the Mediterranean diet is not harmful in patients on hemodialysis with respect to mortality risk, and that limiting the consumption of fruit and vegetables might even be detrimental due to the deprivation of minerals, vitamins, and plant-derived metabolites with anti-inflammatory and antioxidant properties [49]. This has also been recognized by current guidelines on nutrition in CKD [50]. The use of potassium binding resins may allow for greater consumption of fruit, vegetables, and legumes, although the use of these drugs may be associated with side effects.

The situation might be different in KT recipients. There is evidence that the Mediterranean diet could help reduce the risk of post-transplantation diabetes and metabolic syndrome in this population [14,17], as well as the risk of graft failure and kidney function decline [16]. In Dalmatian KT recipients, adherence to the Mediterranean diet was also associated with greater albumin levels and skeletal muscle mass, indicating better nutritional status [51]. In our cohort, the median BMI was significantly greater in KT recipients than in participants on dialysis. Consistently, more participants reported weight gain among KT recipients. This is a well-known issue, which affects a large proportion of KT recipients. Risk factors for post-transplantation weight gain are several, including female sex, pre-transplant body weight [52], younger age, black ethnicity, lower socioeconomic status, diabetes mellitus, acute rejection, steroids, and antidepressants [53]. Dietary regimens based on the principles of the Mediterranean diet might help maintain a healthy weight [54], and existing barriers to the implementation of the Mediterranean diet in the management of KT recipients should be overcome [55]. Recently, practical suggestions for implementing the principles of the Mediterranean diet in renal nutrition have been published by Perez-Torres and colleagues [56]. These address specific food categories (e.g., fruits, vegetables, legumes, cereals, fish…), suggesting the frequency of consumption for each. As an example, the authors recommend 2–3 servings of legumes per week for patients on hemodialysis, whereas a minimum of 4 servings per week is recommended for KT recipients.

It should be acknowledged that despite the strong rationale supporting fewer restrictions regarding fruit, vegetables, and legumes, we did not find an association between adherence to the Mediterranean diet or the reduced consumption of these foods and the presence of comorbidities, including obesity, diabetes, dyslipidemia, hypertension, and vascular disease. On the other hand, we found that excess consumption of dairy products and alcohol and low intake of olive oil were significantly associated with diabetes. Low consumption of olive oil, as well as excess consumption of meat, was associated with obesity. Larger studies are necessary to investigate the dietary habits of individuals on renal replacement therapy, and how these can affect their comorbidities.

Limitations of the present study include its cross-sectional nature, which does not allow us to establish the direction of the associations described here. We were also unable to rule out the possibility that multiple entries from the same individual were included in the analysis as, to keep the survey completely anonymous, we did not collect information (e.g., IP address, cookies) that could be used to identify duplicate entries. The use of self-reported data, particularly height and weight to calculate BMI, is also a limitation of the present study; BMI values based on self-reported anthropometrics tend to overestimate BMI values at the low end of the BMI scale and underestimate BMI values at the high end [57]. Furthermore, we did not investigate alterations in taste perception, which is another factor that might contribute to poor diet. Dysgeusia is common among patients with CKD, particularly those on hemodialysis, and can be due to nutritional deficiencies, dry mouth brought on by water loss, diabetes neuropathy, side effects of drugs, and uremia [58,59]. Strengths of this study include the use of a questionnaire validated in the Italian population [19], the assessment of dietary restrictions, and the involvement of the main national associations of patients with polycystic kidney disease or those on renal replacement therapy, which we believe helped increase interest and awareness of the Mediterranean diet among patients. 

## 5. Conclusions

Adherence to the Mediterranean diet is generally low among individuals on renal replacement therapy; this, at least in part, is due to dietary restrictions, which were significantly more common among participants on dialysis. Participants reported very low rates of fruit, vegetable, and legume consumption. Renal nutrition is complex, and it might be that a “modified” Mediterranean diet, including plant-based proteins, is a good compromise between “too strict” and “no” rules. In light of using food as a therapeutic tool, there is a need for registered dieticians specialized in renal nutrition and of large, well-designed studies addressing these questions.

## Figures and Tables

**Figure 1 ijerph-20-04040-f001:**
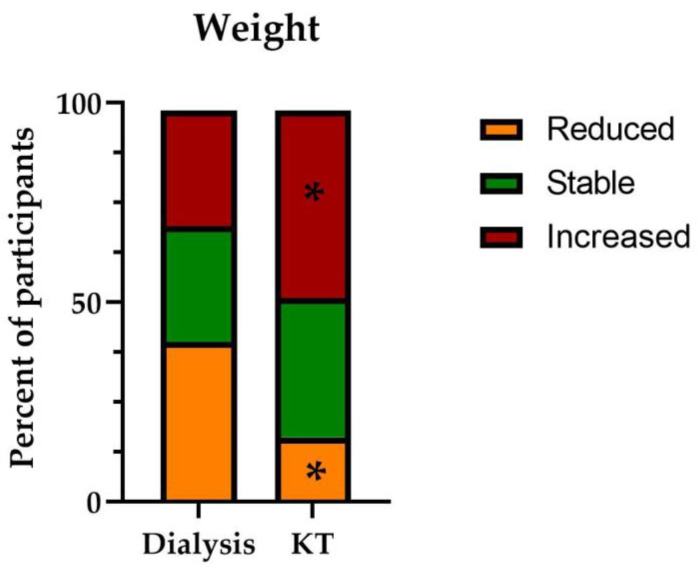
Weight change since the start of dialysis or KT. * *p* < 0.05.

**Figure 2 ijerph-20-04040-f002:**
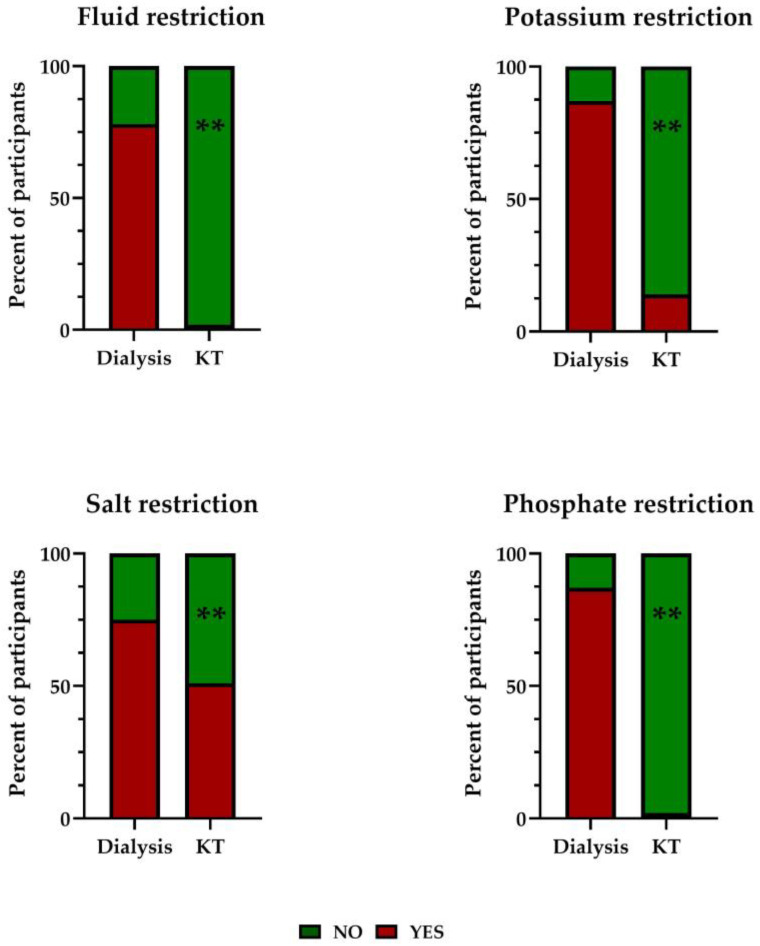
Proportion of participants reporting restrictions in fluid, potassium, salt, or phosphate intake. ** *p* < 0.001 vs. the dialysis group. KT, kidney transplant.

**Figure 3 ijerph-20-04040-f003:**
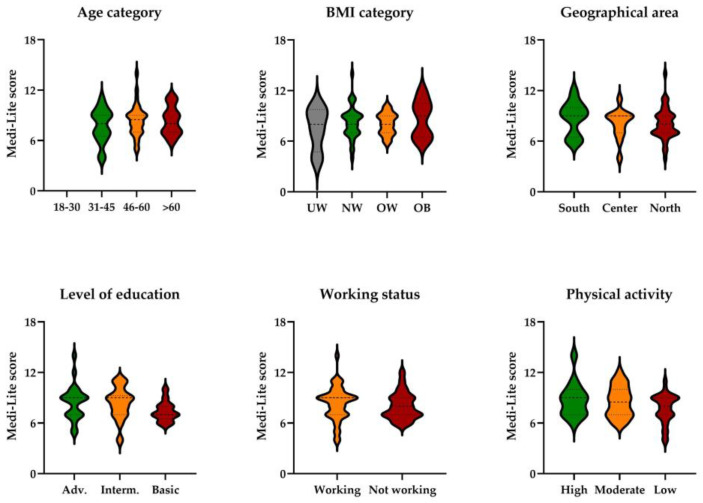
MEDI-LITE score by age category, body mass index (BMI) category, geographical area, level of education, working status, and physical activity level in participants on dialysis. UW, underweight; NW, normal weigh; OW, overweight; OB, obesity; Adv, advanced; Interm., intermediate.

**Figure 4 ijerph-20-04040-f004:**
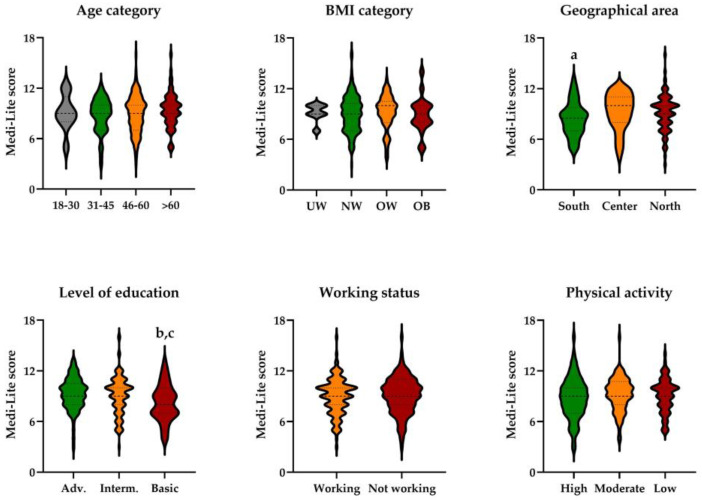
MEDI-LITE score by age category, body mass index (BMI) category, geographical area, level of education, working status and physical activity level in KT recipients. UW, underweight; NW, normal weigh; OW, overweight; OB, obesity; Adv, advanced; Interm., intermediate. ^a^
*p* = 0.031 vs. central Italy; ^b^
*p* = 0.016 vs. intermediate; ^c^
*p* = 0.006 vs. advanced level of education.

**Figure 5 ijerph-20-04040-f005:**
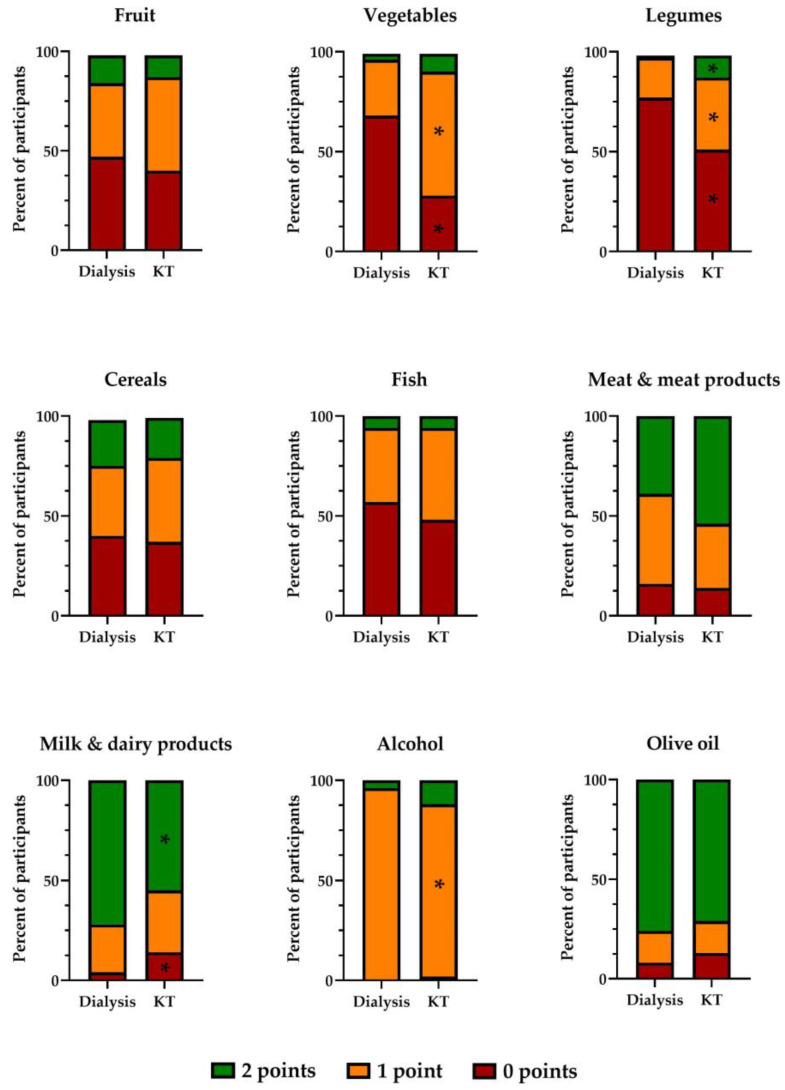
Proportion of participants reporting optimal (2 points), intermediate (1 point), or inadequate (0 points) intake of each food category included in the MEDI-LITE questionnaire. * *p* < 0.05 vs. the corresponding category in the dialysis group.

**Table 1 ijerph-20-04040-t001:** Participant characteristics.

Variable	Kidney Transplant	Dialysis	*p*-Value
Age (years)			0.497
18–30	9 (3.5)	1 (1.5)	
31–45	46 (18.0)	17 (25.4)	
46–60	133 (52.2)	32 (47.8)	
>60	67 (26.3)	17 (25.4)	
Sex (female), *n* (%)	112 (43.9)	26 (38.8)	0.490
BMI category			0.457
- underweight	7 (2.7)	4 (6.0)	
- normal weight	154 (60.4)	38 (56.7)	
- overweight	69 (27.1)	16 (23.9)	
- obesity	25 (9.8)	9 (13.4)	
Active smoker, *n* (%)	17 (6.7)	5 (7.5)	0.959
Geographical area, *n* (%)			0.303
- Northern Italy	181 (71.0)	41 (61.2)	
- Central Italy	38 (14.9)	13 (19.4)	
- Southern Italy	36 (14.1)	13 (19.4)	
Level of education ^1^, *n* (%)			
- Basic	39 (15.3)	13 (19.4)	
- Intermediate	130 (51.0)	30 (44.8)	
- Advanced	85 (33.3)	24 (35.8)	
Worker, *n* (%)	132 (51.8)	42 (62.7)	0.130
Comorbidities, *n* (%)			
- Hypertension	151 (59.2)	33 (49.3)	0.166
- Diabetes	50 (19.6)	11 (16.4)	0.604
- Dyslipidemia	67 (26.3)	13 (19.4)	0.270
- Vascular disease ^2^	20 (7.8)	5 (7.5)	1.000

^1^ Aggregate levels of education according to the International Standard Classification of Education: Basic: primary education or lower secondary education; Intermediate: upper secondary education or post-secondary non-tertiary education; Advanced: short-cycle tertiary education, bachelor’s or equivalent level, master’s or equivalent level, or doctoral or equivalent level [25]. ^2^ Includes cardiovascular and cerebrovascular disease. BMI, body mass index; KT, kidney transplant.

**Table 2 ijerph-20-04040-t002:** Multivariable logistic regression for inadequate adherence to the Mediterranean diet in the whole population, including level of education and dialysis status.

Variable	Odds Ratio	95% CI	*p*
Level of education			
- Advanced (Ref.)	-	-	-
- Intermediate	0.970	0.585; 1.608	0.907
- Basic	2.808	1.285; 6.444	0.010
Dialysis	3.326	1.716; 6.444	<0.001

Adjusted for age and sex. All VIFs were <5.

**Table 3 ijerph-20-04040-t003:** Multivariable logistic regression for inadequate adherence to the Mediterranean diet in the whole population, including level of education and dietary restrictions.

Variable	Odds Ratio	95% C.I.	*p*
Level of education			
- Advanced (Ref.)	-	-	-
- Intermediate	0.997	0.997; 0.599	0.990
- Basic	2.932	1.321; 6.507	0.008
Fluid restriction	3.219	1.126; 9.199	0.029
Potassium restriction	0.971	0.478; 1.973	0.935
Phosphate restriction	1.541	0.573; 4.148	0.391

Adjusted for age and sex. All VIFs were <5.

## Data Availability

The data presented in this study are available on request from the corresponding author.

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
