# Peer review of "Adherence to Mediterranean Diet in Individuals on Renal Replacement Therapy"

_ijerph, 2023, doi:10.3390/ijerph20054040_

Round 1

Reviewer 1 Report

I think more explanation in the result section will be helpful.

Author Response

I and the other authors would like to thank you for the time and effort spent in reviewing our work. We have addressed the comments and extensively revised the manuscript accordingly (edits are marked up in the revised file). Thanks to your insightful suggestions, the quality of the manuscript has improved and we hope that our work will be deemed suitable for publication in the International Journal of Environmental Research and Public Health.

Following is a point-by-point response to the comments (in blue). Line numbers refer to the version of the manuscript with tracked changes (in blue).

Reviewer 1

I think more explanation in the result section will be helpful.

We thank the Reviewer for this suggestion. To better clarify the study flow, we now specify that 333 potential participants accessed the informed consent page, of whom 322 consented to participate (page 3, lines 133-134). We have also added further explanation of the International Standard of Classification of Education in the legend to Table 1 (page 5, lines 153-156).

Reviewer 2 Report

I enjoyed reading this paper, the content is of great interest to dietitians working in the area. The authors should be congratulated on the quality of the presentation. I have some minor suggestions and comments. It is not clear why the authors chose to explore the impact of the Geographical area. Is there known to be a variation in economic status and or consumption of F and V across Italy. This would assist in translation to other countries Additionally greater emphasis on the findings of the impact of education on the consumption of the Med Diet could be made and linked with the literature could be made. Also did the authors consider the potential  impact of taste changes particularly those on dialysis? The limitation of self- report of diet but especially height and weight should be referred to.  One minor typo found in reference 21 Available 

Author Response

I and the other authors would like to thank you for the time and effort spent in reviewing our work. We have addressed the comments and extensively revised the manuscript accordingly (edits are marked up in the revised file). Thanks to your insightful suggestions, the quality of the manuscript has improved and we hope that our work will be deemed suitable for publication in the International Journal of Environmental Research and Public Health.

Following is a point-by-point response to the comments (in blue). Line numbers refer to the version of the manuscript with tracked changes (in blue).

Reviewer 2

I enjoyed reading this paper, the content is of great interest to dietitians working in the area. The authors should be congratulated on the quality of the presentation. I have some minor suggestions and comments.

We thank the Reviewer for her/his comment.

  1. It is not clear why the authors chose to explore the impact of the Geographical area. Is there known to be a variation in economic status and or consumption of F and V across Italy. This would assist in translation to other countries.

We thank the Reviewer for this insightful suggestion. In the introduction, we have expanded the background on the Mediterranean Diet, now mentioning factors that have been shown to be associated with adherence to it, including the impact of the Geographical area (page 1, line 45 and page 3, lines 46-54). In Italy, the consumption of vegetables and the adherence to the Mediterranean Diet was in fact shown to be lowest in Northern regions (Benedetti, Laureti et al. 2018). We have also added a comment on the discrepancy between the lack of association with a Geographical area in our study and previous studies showing a lower adherence to the Mediterranean Diet and consumption of fruit and vegetables  in people from Northern Italy (Correa Leite, Nicolosi et al. 2003, Benedetti, Laureti et al. 2018) in the Discussion (page 11, lines 307-311).

  1. Additionally greater emphasis on the findings of the impact of education on the consumption of the Med Diet could be made and linked with the literature could be made.

In the Discussion, we now stress the importance of implementing educational programmes to increase nutrition literacy and adherence to the Mediterranean Diet, also quoting relevant literature (page 11, lines 299-306).

  1. Also did the authors consider the potential  impact of taste changes particularly those on dialysis?

Unfortunately, we did not assess alteration in taste perception. We now acknowledge this limitation at the end of the Discussion (page 14, lines 455-458). 

  1. The limitation of self- report of diet but especially height and weight should be referred to.

We have added this limitation at the end of the Discussion (page 14, lines 451-454).

  1. One minor typo found in reference 21

Thank you for spotting this, we have corrected it (now reference #25).

Reviewer 3 Report

Please see the attached document with suggestions for revisions.

Author Response

I and the other authors would like to thank you for the time and effort spent in reviewing our work. We have addressed the comments and extensively revised the manuscript accordingly (edits are marked up in the revised file). Thanks to your insightful suggestions, the quality of the manuscript has improved and we hope that our work will be deemed suitable for publication in the International Journal of Environmental Research and Public Health.

Following is a point-by-point response to the comments (in blue). Line numbers refer to the version of the manuscript with tracked changes (in blue).

Reviewer 3

  1. Abstract. Lines 24-25. In this last sentence, there should be a mention of strategies to improve adherence to diets as opposed to or in addition to the quality of the diet. This should be a shared responsibility between registered dietitians, physicians, and the patient.

We thank the Reviewer for this suggestion. We have changed the last sentences of the Abstract accordingly (page 1, lines 24-27).

  1. Introduction. The introduction is very short and should be expanded. lt would help to provide more information about what types of diets are currently recommended by healthcare professionals far dialysis patients or kidney transplant patients and the success of those diets compared to the Mediterranean diet (MD).

We thank the reviewer for this suggestion. Unfortunately, due to lack of randomized clinical trials, currently available guidelines provide general recommendations on energy, sodium, potassium and phosphorus intake and suggest encouraging diets meeting the recommended dietary allowance for adequate intake for all vitamins and minerals (Ikizler, Burrowes et al. 2020), but do not recommend specific dietary patterns. It is suggested, however, that adults with CKD stage 1-5 not on dialysis or post-transplantation follow a Mediterranean Diet pattern to improve lipid profile (Ikizler, Burrowes et al. 2020).  We have added this information to the introduction, along with more details on the Mediterranean Diet (page 1, lines 32-38 and 45; page 2, lines 46-54, lines 56-59).

  1. Early in the introduction you mention a "fear" of the MD. Please clarify if this is fear from the patients or medical professionals. Also consider adding your hypothesis.

Thank you for spotting this imprecision. We have rephrased the sentence (page 1, lines 38-42) to explain that physicians may fear the use of the Mediterranean Diet due to its high content in fruits, vegetables, and legumes, which may contain relevant amounts of potassium and phosphate, and have added a reference to support the concept that dietary restrictions may sometimes be overzealous, leading to deficiencies in important nutrients and poor nutritional status (Kalantar-Zadeh, Tortorici et al. 2015).

  1. Materials and Methods. Please clarify if the participants were asked to follow a MD for a given period of time, or if this was a general survey to examine who might be following the diet by chance. Although this may have been explained in the study design previously published (citation 14), this information is needed here to understand the expectations of the participants. lf they were not asked to follow a MD, then adherence to a diet they were never assigned is a moot point.

This was an observational study to capture the dietary habits of patients on renal replacement therapy in “real life”. Therefore, participants were not asked to follow a Mediterranean Diet for a given period of time. We have now provided more details on the study design at the beginning of the Materials and Methods Section (page 2, lines 73-75). We agree that assessing the adherence to a diet participants were never assigned might be a moot point. However, the same holds true for the huge number of published studies assessing dietary habits in other cohorts. It should also be acknowledged that the Mediterranean Diet is a common eating pattern found in Greece, Italy, Spain, and other countries in the Mediterranean basin, and represents an Intangible Cultural Heritage of these countries (Bonaccio, Iacoviello et al. 2022). As such, it is reasonable to assume that the majority of the population from these countries is aware of the general principles of the Mediterranean Diet.

  1. Results. The results are extensive and presented well. A few suggestions:

  1. Table 1: The hyphens/dashes as they currently are in Table 1 are a bit disordered.

Thank you for pointing out this issue, the formatting has been corrected

  1. Table 2 and 3: Again, the hyphens/dashes could be adjusted to better emphasize the subcategories.

Thank you for pointing out this issue, the formatting has been corrected

  1. Discussion. Overall, this section is well-written and interesting. One sentence specifically needs adjusting - line 327 - "Things might be different in KT recipients". The word "things" could be replaced with "The situation" or "The dietary recommendations" or "The physiological response to a MD...".

Thanks for the suggestion, The word "things" has been replaced with "The situation", as suggested (page 13, line 417).

  1. Additionally, it would help to have more discussion about the significance of the results you reported for the individual food groups, providing a better connection between the results and discussion section.

Thanks for this suggestion. We have expanded the discussion on specific food groups, such as dairy products, fish and olive oil (page 12, lines 361-379 and page 13, lines 380-391). Based on our finding of low adherence to the Mediterranean Diet and given the many benefits associated with this pattern, we also provide a reference with practical suggestions for implementing its principles into renal nutrition (Perez-Torres, Caverni-Munoz et al. 2022) (page 13, lines 431-433 and page 14, lines 434-436).   

  1. You do mention limitations of your study, but how about strengths? A brief mention of the disadvantages of using a self­reporting questionnaire could also be included.

We have added the strengths of our study (using a questionnaire validated in the Italian population and the involvement of patient associations) (page 14, lines 459-463). We have also added that the use of self-reported data, particularly height and weight to calculate BMI, is another a limitation of our study (page 14, lines 451-454).

  1. Throughout the manuscript, there was an inconsistent use of the Oxford comma. This is a minor grammatical issue that could be reviewed and improved as you make revisions.

Thank you for spotting out this issue, we have reviewed and corrected the English language and the use of the Oxford comma throughout the text.

Reviewer 4 Report

Since the data is self-report, the conclusions need to address this shortcoming

A comparison group would be significantly helpful with interpretation.  Can a group of participants not on renal replacement therapy be used to compare and have a better understanding of those on dialysis or renal transplant and the Mediterranean diet?

Methods

More detail is need regarding the arrival of 322 participants in the study.  Since this was a secondary data analysis, what was the original sample size of those that completed all questionnaires and what were the inclusion and exclusion criteria used to narrow the same to 322?

International standard of classification of education needs further explanation for readers to understand the different levels in table 1

Further definition is need as to what an active worker would entail.  Part-time, full-time seasonal etc.

Discussion

Lines 268-269 state: . It is not surprising that participants on dialysis had the lowest adherence to Mediterranean Diet, which was influenced by dietary restrictions typically adopted in this population”.  It would be helpful to provide a reference or compare to previous literature to make this statement.

Were the participants educated about a Mediterranean diet and asked to follow that diet or were they simply following any diet they preferred?  An explanation of this will be helpful in the method section.

Author Response

I and the other authors would like to thank you for the time and effort spent in reviewing our work. We have addressed the comments and extensively revised the manuscript accordingly (edits are marked up in the revised file). Thanks to your insightful suggestions, the quality of the manuscript has improved and we hope that our work will be deemed suitable for publication in the International Journal of Environmental Research and Public Health.

Following is a point-by-point response to the comments (in blue). Line numbers refer to the version of the manuscript with tracked changes (in blue).

Reviewer 4

  1. Since the data is self-report, the conclusions need to address this shortcoming.

Thanks for this suggestion. At the end of the Discussion, we have added that the use of self-reported data, particularly height and weight to calculate BMI, is another a limitation of our study (page 14, lines 451-454).

  1. A comparison group would be significantly helpful with interpretation. Can a group of participants not on renal replacement therapy be used to compare and have a better understanding of those on dialysis or renal transplant and the Mediterranean diet?

We thank the Reviewer for raising this point. Our study was specifically focussed on patients on renal replacement therapy, we did not add a control group. No studies assessed the adherence to the Mediterranean Diet using the Medi-Lite questionnaire in patients with CKD 1-5 not on dialysis nor transplanted. However, there are data regarding the general Italian population that could help interpreting our findings. We have added this information to the Discussion (page 11, lines 284-291). Thank you for this suggestion.

Methods

  1. More detail is need regarding the arrival of 322 participants in the study. Since this was a secondary data analysis, what was the original sample size of those that completed all questionnaires and what were the inclusion and exclusion criteria used to narrow the same to 322?

We are sorry about the lack of clarity on the study design. In this secondary analysis, we assessed the adherence to Mediterranean Diet among individuals on dialysis or who received a KT, and factors that influence it. To this aim, we included all participants who consented to participate in the study. In the Methods, we now specify that all patients who consented to participate in the survey were included in the analysis (page 2, lines 81-82). In the Results, we now specify that 333 potential participants accessed the informed consent page, of whom 322 consented to participate (page 3, lines 133-134).

  1. International standard of classification of education needs further explanation for readers to understand the different levels in table 1

Thank you for this suggestion. We have added further explanation of the International Standard of Classification of Education in the legend to Table 1.

  1. Further definition is need as to what an active worker would entail.  Part-time, full-time seasonal à just worker

Thank you for spotting this imprecision. As we were not able to discriminate among work type (part-time, full-time, seasonal…) we changed “active worker” to “worker”, as suggested (page 4, line 148 and Table 1).

Discussion

  1. Lines 268-269 state: It is not surprising that participants on dialysis had the lowest adherence to Mediterranean Diet, which was influenced by dietary restrictions typically adopted in this population”.  It would be helpful to provide a reference or compare to previous literature to make this statement.

We have rephrased the sentence to support the statement that it is not surprising that participants on dialysis had the lowest adherence to Mediterranean Diet (page 11, lines 319-325). This diet is rich in fruits, vegetables, and legumes that may be a relevant source of potassium and phosphate. Guidelines state that it is reasonable to adjust dietary intake of these minerals to maintain serum levels within the normal range (Ikizler, Burrowes et al. 2020). This possibly explains why, in our study, adherence to the Mediterranean Diet was influenced by dietary restrictions typically adopted in this population. This issue is further discussed in the following paragraphs (e.g., page 12, lines 329-33, lines 356-358, lines 366-369). Our finding of low adherence to the Mediterranean Diet is consistent with the few available data on patients on hemodialysis (Saglimbene, Wong et al. 2018) (page 11, lines 312-313).

  1. Were the participants educated about a Mediterranean diet and asked to follow that diet or were they simply following any diet they preferred?  An explanation of this will be helpful in the method section.

This was an observational study to capture the dietary habits of patients on renal replacement therapy in “real life”. Therefore, participants were not educated about a Mediterranean diet and asked to follow it. We have now provided more details on the study design at the beginning of the Materials and Methods Section (page 2, lines 73-75).. We acknowledge that assessing the adherence to a diet participants were not educated about might be debatable. However, the same holds true for the huge number of published studies assessing dietary habits in other cohorts. It should also be acknowledged that the Mediterranean Diet is a common eating pattern found in Greece, Italy, Spain, and other countries in the Mediterranean basin, and represents an Intangible Cultural Heritage of these countries (Bonaccio, Iacoviello et al. 2022). As such, it is reasonable to assume that the majority of the population from these countries is aware of the general principles of the Mediterranean Diet.

Round 2

Reviewer 3 Report

The revised version of this manuscript is significantly improved. Thank you for taking the time to make adjustments based on the reviewers' suggestions.